Relation between peripheral blood inflammatory indices and severity of central retinal artery occlusion

Hu Weiwen 1 2
Huang Yikeng 3
Zhou Qiong zqndyfy@163.com 2
Huang Xionggao hxg_eye@163.com 1 4
1 Department of Ophthalmology, The First Affiliated Hospital of Hainan Medical University , Haikou , Hainan , China
2 Department of Ophthalmology, The First Affiliated Hospital, Jiangxi Medical College, Nanchang University , Nanchang , Jiangxi , China
3 Department of Ophthalmology, Shanghai General Hospital, Shanghai Jiao Tong University School of Medicine , Shanghai , China
4 Key Laboratory of Emergency and Trauma of Ministry of Education, Department of Emergency Surgery, Key Laboratory of Hainan Trauma and Disaster Rescue, The First Affiliated Hospital, Hainan Medical University , Haikou , Hainan , China
Rocha Joao
Electronic publication date: 2024 Sep 30
Publication date: 2024
Volume: 12
Electronic Location ID: e18129
Received 2024 Apr 17; Accepted 2024 Aug 28
Copyright: ©2024 Hu et al.
Copyright year: 2024
Copyright holder: Hu et al.
License: This is an open access article distributed under the terms of the Creative Commons Attribution License, which permits unrestricted use, distribution, reproduction and adaptation in any medium and for any purpose provided that it is properly attributed. For attribution, the original author(s), title, publication source (PeerJ) and either DOI or URL of the article must be cited.
License URL: https://creativecommons.org/licenses/by/4.0/

Keywords: Central retinal artery occlusion, Severity, Neutrophil-to-lymphocyte ratio, Monocyte-to-high-density lipoprotein cholesterol ratio, Systemic immune-inflammation index

Funding: The National Natural Science Foundation of China 81860172 82160199 82260211 81460092 Hainan Provincial Natural Science Foundation of China No.: 821RC1126 The Central Government Guides Local Science and Technology Development Foundation 20211ZDG02003 The Key research and development project in Jiangxi Province 20203BBG73058 20192BBGL70033 The Chinese medicine science and technology project in Jiangxi province 2020A0166 The study was supported by grants from the National Natural Science Foundation of China (No.: 81860172, Xionggao Huang, No.: 82160199, Xionggao Huang, No.: 82260211, Qiong Zhou and 81460092, Qiong Zhou), the Hainan Provincial Natural Science Foundation of China (No.: 821RC1126, Xionggao Huang), the Central Government Guides Local Science and Technology Development Foundation (No.: 20211ZDG02003, Qiong Zhou), the Key research and development project in Jiangxi Province (No.: 20203BBG73058, Qiong Zhou and 20192BBGL70033, Qiong Zhou), and the Chinese medicine science and technology project in Jiangxi province (No.: 2020A0166, Qiong Zhou). The funders had no role in study design, data collection and analysis, decision to publish, or preparation of the manuscript.

==============================
Background

Central retinal artery occlusion (CRAO) has been identified as an acute emergency resulting in vision loss, with its pathogenesis potentially involving systemic inflammation and abnormal lipid metabolism. Over recent years, it has been established that peripheral blood inflammatory indices, including the neutrophil-to-lymphocyte ratio (NLR), the systemic immunoinflammatory index (SII), and the monocyte-to-high-density lipoprotein ratio (MHR), play significant roles in assessing systemic inflammation and lipid metabolism. However, the role of these indices in assessing the severity of CRAO has rarely been explored. This study aimd to investigate the relationship between these inflammatory indices and the severity of CRAO.

Methods

This was a retrospective clinical study with a total of 49 CRAO patients and 50 age- and sex-matched controls involved. The patients with CRAO were divided into three groups (13 with incomplete CRAO, 16 with subtotal CRAO and 20 with total CRAO). Data were compared across these groups, and additionally, correlation analysis, restricted cubic spline plots, and receiver operating characteristic curve analysis were performed.

Results

The values of NLR, SII and MHR were significantly higher in the CRAO group compared to controls (NLR: 2.49(1.71,3.44) vs 1.60(1.24,1.97), P<0.001; SII: 606.46(410.25,864.35) vs 403.91(332.90,524.31), P=0.001; MHR: 0.33(0.26,0.44) vs 0.25(0.21,0.34), P<0.001). MHR was also significantly higher in total CRAO than in incomplete CRAO and subtotal CRAO (0.41(0.32,0.60) vs 0.29(0.21,0.43), P=0.036; 0.41(0.32,0.60) vs 0.29(0.23,0.38), P=0.017). Significant positive associations were found between MHR, NLR, SII and both the incidence (all P<0.001) and severity (P<0.001, P<0.001, P=0.003, respectively) of CRAO. MHR had a linear relationship with both the occurrence and severity of CRAO (P-overall=0.013, P-non-linear=0.427 and P-overall=0.013, P-non-linear=0.825). Combining MHR and NLR significantly improved diagnostic efficacy for CRAO and total CRAO, with area under the curve of 0.816 and 0.827, respectively, compared to using MHR alone (0.705 and 0.816).

Conclusion

Elevated levels of peripheral blood NLR, SII, and MHR are positively associated with CRAO incidence, highlighting their potential as early predictive markers. The combined NLR and MHR index further enhances diagnostic accuracy and may facilitate timely assessment of CRAO severity by ophthalmologists and internists.

Introduction

Acute central retinal artery occlusion (CRAO) is typically associated with retinal ischaemia and compromises the processing of visual information, constituting a classical etiology for acute, painless vision loss. The age- and sex-adjusted annual incidence of CRAO ranges between 1.8 to 10.1 per 100,000 and escalates with advancing age (Park et al., 2024). Studies have shown that visual acuity in patients with CRAO can range from 20/25 to complete absence of light perception, with over 92% eventually achieving vision of counting finger or worse (Chan et al., 2021). Most patients fail to regain their vision despite treatment. Prognostic factors influencing the outcome of CRAO include the time to treatment, and the severity (Dumitrascu, Newman & Biousse, 2020). Schmidt, Schulte-Mönting & Schumacher (2002) were the first to divide CRAO into incomplete, subtotal and total types, finding that patients with incomplete CRAO experienced the greatest improvement in vision, whereas those with total CRAO saw less significant benefits from treatment. Hence, timely identification of CRAO and assessment of its severity can guide treatment decisions, forecast prognosis, and facilitate the recognition and management of systemic health issues associated with CRAO.

The diagnosis of CRAO is fundamentally reliant on distinct symptomatic manifestations, and fundoscopic evidence of grey-white retinal oedema, accompanied by a cherry-red spot at the macula. Fluorescein angiography reveals a protracted arm-to-retina circulation time and occlusion within the retinal arteries. Similarly, optical coherence tomography exhibits increased reflectivity within the inner retinal layers (Matthe, Eulitz & Furashova, 2020). Despite well-defined diagnostic criteria for CRAO, certain limitations persist, including dependency on the expertise of ophthalmologists and the inadequate capacity to identify atypically presenting cases promptly. The aetiology of CRAO is acknowledged as multifaceted, involving both ocular and systemic factors. Inflammation and abnormalities in lipid metabolism, leading to vascular endothelial dysfunction, intravascular thrombosis, and atherosclerosis, have been identified as significant contributors to the development of CRAO (Dziedzic et al., 2023; Vujosevic et al., 2023). Thus, biomarkers that assess systemic inflammation and lipid metabolism could potentially serve as predictive and diagnostic tools for CRAO.

Neutrophils, lymphocytes and monocytes, integral to the body’ defence system, are involved in vascular endothelial injury, oxidative stress and inflammation (Nost et al., 2021). The neutrophil-to-lymphocyte ratio (NLR) is derived from the quotient of neutrophils to lymphocytes in peripheral blood, serving as a widely acknowledged marker of systemic inflammation. Elevated NLR values correlate with adverse cardiovascular prognosis and heightened mortality, thus highlighting its relevance in vascular inflammation and thrombosis (Dong et al., 2023). The systemic immune-inflammation index (SII), incorporating neutrophil, lymphocyte, and platelet counts, provides a more comprehensive evaluation of an individual’ immune-inflammatory status. Increased SII values denote enhanced immune and inflammatory responses, implicated in the pathogenesis of diverse cardiovascular and metabolic diseases (Kearney, Alsharqi & Kirby, 2022). High-density lipoprotein cholesterol (HDL-C) is noted for its beneficial role, mitigating monocyte activity and countering low-density lipoprotein cholesterol (LDL-C) through its anti-inflammatory, anti-thrombotic and circulation-improving protective role (Hwang et al., 2022). The monocytes to HDL-C ratio (MHR) acts as a dual indicator of inflammatory status and lipid metabolism (Zhou et al., 2021). Monocytes play a pivotal role in inflammation and atherosclerosis, while HDL-C is renowned for its protective effect against cardiovascular diseases. An elevated MHR signifies a pro-inflammatory and pro-atherogenic states, potentially contributing to vascular events such as acute coronary syndrome, acute ischemic stroke, and atherosclerosis (Sun et al., 2020; Xu et al., 2023; Zhou et al., 2021). Recent studies have identified NLR, SII and MHR as emerging markers for inflammation associated with CRAO, underscoring their potential as novel biomarkers (Elbeyli et al., 2022; Guven & Kilic, 2021; Pinna et al., 2021; Qin et al., 2022; Zhang, Xing & Deng, 2023). Nevertheless, the paucity of extensive research generates uncertainties regarding the precise traits of blood biomarkers across different CRAO severity levels. Consequently, this study aims to scrutinise the variations in blood biomarkers associated with inflammation in CRAO and to thoroughly investigate the correlation between these biomarkers and CRAO, segmented by its severity.

Material and Methods

This study received approval from the Ethics Committee at the First Affiliated Hospital of Hainan Medical University (Approval No.: 2022087), ensuring adherence to the ethical guidelines outlined in the Declaration of Helsinki. We included patients diagnosed with non-arteritic CRAO by the Ophthalmology Department of the same institution from January 2015 through January 2024 in this retrospective clinical analysis. The patients were informed about the study, and written informed consent was obtained from all participants.

Exclusion criteria encompassed patients presenting with other ocular conditions, including various types of glaucoma, uveitis, retinal detachment, and optic neuritis. Additionally, a history of ocular surgery, such as vitrectomy and cataract phacoemulsification, warranted exclusion. Patients recently receiving anti-coagulants, anti-platelet therapies, systemic corticosteroids, or anti-inflammatory medications were also excluded. Furthermore, individuals suffering from hepatic or renal failure, or immune system disorders, such as systemic lupus erythematosus and vasculitis, were not considered for inclusion.

Each participant underwent a comprehensive ophthalmic assessment. This evaluation included the measurement of the best corrected visual acuity, intraocular pressure, examination of the dilated pupil, and a slit lamp fundus examination. Advanced diagnostic imaging techniques were also employed, including optical coherence tomography (Carl Zeiss, HD-OCT, Germany) and fluorescein fundus angiography (Topcon, TRC-50DX, Japan).

In this study, the CRAO grading was adopted from the literature (Furashova & Matthe, 2017; Matthe, Eulitz & Furashova, 2020), and the specific characteristics of the grading are detailed in Table 1. Specifically, patients afflicted with CRAO were classified based on the duration of retinal artery filling, the presence of cherry-red spots, retinal thickness, and morphological features as incomplete, subtotal, or total (Fig. 1). Owing to the lack of a universally acknowledged definition for the timing of retinal artery trunk filling across different severities of occlusion, and considering that the filling times in unaffected individuals range from 8 to 15 s, the following delineations were established as a reference: delays of up to 15 s were classified as mild, up to 30 s as moderate, and beyond 30 s as severe. In cases of incomplete CRAO, characteristics included a mild delay in retinal artery trunk filling, absence of pronounced cherry-red spots, enhanced reflection from the inner retinal layers, no substantial increase in retinal thickness, and preservation of a clear retinal stratification (Figs. 1D–1F). Subtotal CRAO was identified by a moderate delay in artery filling, observable cherry-red spots, increased inner retinal reflections, thickening of the retina, and some disruption to retinal structure, though the layering remained discernible (Figs. 1G–1I). Total CRAO was characterised by significant delays exceeding 30 s in artery filling, evident cherry-red spots, augmented inner retinal reflexes, considerable retinal thickening, edema, and extensive disruption of the retinal structure and clarity (Figs. 1J–1L). The classification was conducted by two researchers, with the final grouping subsequently validated and determined by two senior chief ophthalmologists.

Table 1 Characteristics of optical coherence tomography and fluorescein fundus angiography in different grades of central retinal artery occlusion.

Grade
of
central retinal artery occlusion	Optical coherence tomography	Cherry-red spot of the macula	Blood flow revealed	
	Hyperreflectivity of inner retinal layers	Thickening of retinal layers	Loss of organized layer structure	Intact layer-by-layer retinal structure			
Incomplete	Mild enhancement	Not obvious or mild thickening	Not	Intact	Not obvious	Mild delay	
Subtotal	Moderate enhancement	Moderate thickening	Not or mild loss	Partial disruption	Obvious	Moderate delay	
Total	Severe enhancement	Severe thickening	Obvious	Complete disruption	Obvious	Severe delay	
Notes.

Normal retinal artery trunk filling time is typically observed within 8–15 s. In this study, delay categories for central retinal artery occlusion are preliminarily set as: mild within 15 s, moderate within 30 s, and severe exceeding 30 s.

Figure 1 Photographic images of the color fundus (left), fluorescein angiography (center), and optical coherence tomography (right) displayed to show the ophthalmic imaging features for normal control, incomplete CRAO, subtotal CRAO, and total CRAO groups.

The normal healthy retina with clear fundus vascular structures, an intact optic disc (A), rapid filling of retinal arteries (B), and high retinal morphology recognition. (C) Incomplete CRAO with mild retinal whitening and narrowing of retinal arterioles without obvious cherry-red patches (D), retinal artery trunk fluorescence completely filled approximately 28.3 s after fluorescent agent injection (E), and mild enhancement of the nasal inner retinal reflexes without obvious retinal thickness thickening and structural disturbances. (F) Subtotal CRAO with pronounced retinal whitening, arterial narrowing, a cherry-red spot in the macula (G), nearly filled retinal artery trunk at 39.5 s (H), markedly enhanced inner retinal reflections, and a widened outer nuclear layer resulting in significant retinal thickening, yet recognisable structures. (I) Total CRAO with diffuse retinal clouding and whitening (J), incompletely filled retinal artery trunk at 46.8 s post-fluorescent injection (K), significant hyperreflectivity in the inner retina, retinal bulging, and structural disturbances (L).

Elbow vein blood is collected early in the morning on the second day after admission for routine blood tests, with patients typically required to fast for 12 h. The levels of white blood cells, neutrophils, lymphocytes, monocytes, and platelets were determined using an automated hematology analyzer (XN-9000, SYSMEX, Japan). Additionally, creatinine, blood urea nitrogen, total cholesterol, HDL-C, and LDL-C were measured with a fully automated biochemical analyzer (SIEMENS Aptio™, Munich, Germany). The NLR, MHR, and the PLR were calculated manually. The SII was derived by multiplying the platelet count by the NLR.

Statistical Analysis

Statistical analyses were performed using SPSS software (SPSS, 26.0 Chicago, IL, USA). The normality of continuous variables was assessed using the Shapiro–Wilk test. Descriptive statistics included the mean and standard deviation (mean ±SD), median and interquartile range (M(P25,P75)), and percentages (%). Differences in quantitative values between CRAO patients and control subjects were evaluated using either the Student’s t-test or the Mann–Whitney U test, as appropriate. For comparing metrics across the three study groups, one-way ANOVA or the Kruskal-Wallis H test was applied. Qualitative variables were compared using the chi-square test. To analyse the correlation between various indices and the occurrence and severity of CRAO, the Spearman test was employed. Meanwhile, the association between the MHR, NLR, and SII with both the occurrence and severity of CRAO were are derived from a univariate logistic regression model adjusted using restricted cubic spline curves with three knots at the 10th, 50th, and 90th percentiles. Furthermore, to assess the diagnostic potential of the newly proposed models for CRAO, logistic regression was initially employed to derive proportions from combinations of MHR and NLR, resulting in a novel model termed MHR_NLR. Similarly, a combination of MHR and SII was formulated, denoted as MHR_SII. The predictive accuracy of the NLR, SII, MHR, MHR_NLR, and MHR_SII models was evaluated through receiver operating characteristic curve analysis. The areas under the curve and 95% confidence intervals were calculated for each model. All statistical tests were conducted bivariately, with P values less than 0.05 deemed statistically significant.

Results

During the study period, 69 subjects were initially diagnosed with CRAO. Exclusions were made as follows: six subjects lacked fluorescein fundus angiography; four had undergone vitrectomy surgery; three were receiving anticoagulation therapy for valvular heart disease; two had glaucoma; two were diagnosed with systemic lupus erythematosus; two had chronic obstructive pulmonary disease; and one had diabetic renal failure. After these exclusions, 49 patients with confirmed CRAO and 50 age- and sex-matched healthy controls were enrolled in the study.

Table 2 summarizes baseline characteristics, including NLR, SII, MHR, and other blood parameters for both groups. The mean age was 61.55 ± 13.53 years for the CRAO group and 61.06 ±11.45 years for the control groups (P = 0.464). The two groups were close in regard to gender (P = 0.104) and body mass index (P = 0.814). Additionally, no statistical significance was found regarding smoking status (P = 0.755) or the presence of hypertension (P = 0.605) between the two groups.

Table 2 Clinical features and peripheral hematological parameters of CRAO and control groups.

	CRAO(n:49)	Controls(n:50)	P value*	
	mean ±SD	mean ±SD		
Gender (male-%)	36(73.5)	29(58.0)	0.10	
LDL-C (mmol/L)	3.19 ± 1.17	2.67 ± 0.67	0.008	
HDL-C (mmol/L)	1.40 ± 0.41	1.55 ± 0.28	0.032	
Total cholesterol (mmol/L)	5.35 ± 1.52	4.51 ± 0.91	0.001	
	Median (IQR)	Median (IQR)	P value#	
Age (years)	63.00(52.50,72.50)	58.00(52.75,67.50)	0.46	
White blood cell (109/L)	7.40(6.09,9.49)	6.26(5.45,7.87)	0.001	
Neutrophil (109/L)	4.73(3.83,5.88)	3.70(2.90,4.44)	<0.001	
Lymphocyte (109/L)	1.83(1.51,2.34)	2.20(1.81,2.67)	0.06	
Monocyte (109/L)	0.48(0.36,0.60)	0.39(0.34,0.48)	0.008	
Platelet (109/L)	254.0(191.0,290.5)	260(227.7,306.7)	0.18	
NLR	2.49(1.71,3.44)	1.60(1.24,1.97)	<0.001	
PLR	120.3(100.3,152.8)	114.0(98.25,144.1)	0.42	
MLR	0.25(0.19,0.30)	0.18(0.14,0.24)	0.001	
MHR (109/mmol)	0.33(0.26,0.44)	0.25(0.21,0.34)	<0.001	
SII	606.46(410.25,864.35)	403.91(332.90,524.31)	0.001	
Notes.

Abbreviations HDL-C high density lipoprotein cholesterol

LDL-C low density lipoprotein cholesterol

NLR neutrophil-to-lymphocyte ratio

PLR platelet-to-lymphocyte ratio

MLR monocyte-to-lymphocyte ratio

MHR monocyte to HDL-C ratio

SD standard deviation

IQR interquartile range (M(P25,P75))

* Student’s t-test or chi-square test was used.

# Mann–Whitney U test was used.

There were significantly higher white blood cells, neutrophils, monocytes, LDL-C and total cholesterol in CRAO group than in controls (P = 0.001, <0.001, 0.008, 0.008 and 0.001, respectively). Conversely, HDL-C levels were significantly lower in the CRAO group than controls (P = 0.032). The NLR, SII and MHR were also significantly higher in CRAO group than in the control group (P = <0.001, 0.001, <0.001, respectively). There were no statistical differences in lymphocytes, platelet count, blood urea nitrogen, creatinine and PLR between the CRAO and control groups (P = 0.057, 0.175, 0.103, 0.954, 0.417, respectively) (Table 2).

Forty-nine CRAO patients were further divided into three groups: 13 patients with incomplete CRAO, 16 patients with subtotal CRAO and 20 patients with total CRAO. Details of the OCT and FFA features of each type of CRAO were presented in Table 1. The baseline information and hematology parameters for each group of CRAO were similarly detailed in Table 3. The age (P = 0.122) and duration of CRAO (P = 0.050) were not statistically different among three groups. MHR was significantly higher in total CRAO group than incomplete and subtotal (P = 0.036, P = 0.017, respectively).

Table 3 Clinical features and peripheral hematological parameters in different grades of CRAO.

	Incomplete(n:13)	Subtotal(n:16)	Total(n:20)	P value*	
	Mean ±SD	Mean ±SD	Mean ±SD		
HDL-C (mmol/L)	1.54 ± 0.49	1.52 ± 0.37	1.22 ± 0.30	0.028	
Arm-retinal artery filling time (s)	18.47 ± 5.31	34.38 ± 6.96	72.85 ± 22.62	<0.001	
Durationa(hr)	124.23 ± 156.80	41.75 ± 41.07	57.10 ± 61.17	0.05	
	Median (IQR)	Median (IQR)	Median (IQR)	P value#	
Age (years)	66.00(53.50,73.50)	66.00(59.25,76.50)	55.50(47.75,67.00)	0.12	
Neutrophil (109/L)	5.12(4.44,6.61)	4.60(3.76,6.02)	4.62(3.69,5.56)	0.67	
Monocyte (109/L)	0.48(0.23,0.59)	0.42(0.35,0.54)	0.49(0.44,0.67)	0.18	
NLR	2.58(1.99,3.44)	2.88(1.70,3.54)	1.98(1.48,3.17)	0.50	
MLR	0.24(0.14,0.29)	0.24(0.21,0.31)	0.28(0.20,0.31)	0.41	
MHR (109/mmol)
SII	0.29(0.21,0.43)
642.04(525.11,815.03)	0.29(0.23,0.38)
601.85(376.03,919.57)	0.41(0.32,0.60)
543.83(383.19,964.10)	0.007
0.91	
Notes.

Abbreviations HDL-C high density lipoprotein cholesterol

NLR neutrophil-to-lymphocyte ratio

MLR monocyte-to-lymphocyte ratio

MHR monocyte to HDL-C ratio

SD standard deviation

IQR interquartile range [M(P25,P75)]

a Duration:time elapsed from symptom onset to hospital admission.

* One-way ANOVA test was used.

# Kruskal-Wallis H-test was used.

Significant positive associations between MHR, NLR, SII and both the incidence (all P < 0.001) and severity (P < 0.001, P < 0.001, P = 0.003, respectively) of CRAO were indicated by Spearman analysis (Table 4).

Table 4 Correlation analysis of blood parameters with the occurrence and severity of retinal artery occlusion.

Characteristic	Occurrence of RAO	Severity of RAO	
	r s	P	r s	P	
Age	0.074	0.47	0.01	0.92	
Gender	−0.163	0.12	−0.154	0.13	
White blood cell	0.325	0.001	0.3	0.003	
Neutrophil	0.433	<0.001	0.376	<0.001	
Lymphocyte	−0.192	0.057	−0.16	0.11	
Monocyte	0.27	0.007	0.31	0.002	
Platelet	−0.137	0.18	−0.075	0.46	
Total Cholesterol	0.307	0.002	0.284	0.004	
Triglycerides	0.297	0.003	0.291	0.003	
HDL-C	−0.266	0.008	−0.343	0.001	
LDL-C	0.253	0.011	0.278	0.005	
MHR	0.356	<0.001	0.429	<0.001	
NLR	0.433	<0.001	0.355	<0.001	
PLR	0.082	0.42	0.08	0.43	
MLR	0.349	<0.001	0.367	<0.001	
SII	0.345	<0.001	0.3	0.003	
Notes.

Abbreviations HDL-C high density lipoprotein cholesterol

LDL-C low density lipoprotein cholesterol

NLR neutrophil-to-lymphocyte ratio

PLR platelet-to-lymphocyte ratio

MLR monocyte-to-lymphocyte ratio

MHR monocyte to HDL-C ratio

RAO retinal artery occlusion

In addition, we flexibly modelled the association of MHR, NLR and SII with the occurrence and severity of CRAO using univariate logistic regression models, as detailed in Fig. 2. Statistical analysis demonstrated significant correlations between inflammatory indices (MHR, NLR, and SII) and both the incidence and severity of CRAO. Regarding the incidence of CRAO, significant linear correlations were established across all indices (MHR: P-overall =0.013, NLR: P-overall<0.001, SII: P-overall =0.004), with non-linear relationships in MHR (P-non-linear =0.427) and NLR (P-non-linear =0.226) not reaching statistical significance, indicating a primary linear relationship. Concering CRAO severity, significant correlations were noted (MHR: P-overall =0.013, NLR: P-overall =0.009, SII: P-overall =0.040). However, non-linear relationship were significant for NLR (P-non-linear =0.002) and SII (P-non-linear =0.012), suggesting complex interactions among these markers and the severity of CRAO that do not conform to a straightforward linear progression.

Figure 2 Association of MHR, NLR, and SII with the occurrence and severity of CRAO using restricted cubic spline curves.

(A, C and E) illustrate the association with CRAO occurrence for MHR, NLR, and SII, respectively. (B, D and F) show their association with CRAO severity. Odds ratio (OR, solid red lines) and 95% confidence intervals (95% CI, light red areas) are derived from a univariate logistic regression model adjusted using restricted cubic spline curves with three knots at the 10th, 50th, and 90th percentiles. P-overall values below 0.05 denote statistically significant associations, and P-non-linear values below 0.05 signal non-linear relationships between the variable and the outcome. Histograms indicate the distributions of MHR, NLR, and SII. MHR, monocyte-to-high-density lipoprotein ratio; NLR, neutrophil-to-lymphocyte ratio; SII, systemic immunoinflammatory index; CRAO, central retinal artery occlusion.

The cut-off value of NLR, SII, MHR, MHR_NLR and MHR_SII for CRAO diagnosis were 2.055, 544.280, 0.301, 0.367 and 0.342 respectively, with areas under the curve were 0.750, 0.699, 0.705, 0.816 and 0.765 respectively. Further ROC analysis for different grades of CRAO, we found a cut-off value of MHR_NLR at 0.124 with 90% sensitivity and 62% specificity for total CRAO, and areas under the curve reached 0.827 (Fig. 3 and Table 5).

Figure 3 ROC analysis of diagnostic efficacy for CRAO using MHR, NLR, and SII.

Best cut-off value and sensitivity and specificity results of NLR, SII, MHR, MHR_NLR and MHR_SII in ROC curves for CRAO (A). Best cut-off value and sensitivity and specificity results of NLR, SII, MHR, MHR_NLR and MHR_SII in ROC curves for total CRAO (B). ROC: receiver operating characteristic curve, MHR: monocyte-to-high-density lipoprotein ratio, NLR: neutrophil-to-lymphocyte ratio, SII: systemic immunoinflammatory index, CRAO: central retinal artery occlusion.

Table 5 Summary of ROC curves analysis.

	AUC	YI	Cut-off	Sensitivity	Specificity	95% CI	
For CRAO		
NLR	0.750	0.453	2.055	0.633	0.820	0.653–0.847	
SII	0.699	0.413	544.280	0.633	0.780	0.595–0.803	
MHR	0.705	0.333	0.301	0.653	0.680	0.603–0.808	
MHR_NLR	0.816	0.478	0.367	0.878	0.600	0.734–0.897	
MHR_SII	0.765	0.418	0.342	0.878	0.540	0.673–0.857	
For total CRAO		
NLR	0.568	0.206	1.690	0.750	0.456	0.419–0.718	
SII	0.591	0.199	1045.608	0.250	0.949	0.446–0.736	
MHR	0.816	0.521	0.319	0.850	0.671	0.719–0.913	
MHR_NLR	0.827	0.520	0.124	0.900	0.620	0.729–0.924	
MHR_SII	0.811	0.482	0.114	0.900	0.592	0.711–0.912	

Discussion

CRAO is globally recognized as a severe retinal vascular condition, potentially leading to irreversible vision loss if not addressed promptly. Although the pathobiology of CRAO is multifactorial and its pathogenesis remains elusive, inflammation, abnormal lipid metabolism, and thrombosis are widely acknowledged as pivotal factors in its development. Recent studies have identified the NLR, SII, and MHR as innovative markers of inflammation, abnormal lipid metabolism and thrombosis. In this study, increases in serum NLR, SII and MHR levels were observed among patients with CRAO, corroborating finding from previous studies and underscoring the utility of these markers across diverse ethnicities. Moreover, it was found that these blood biomarkers do not entirely correlate with the severity of CRAO; only an elevation in MHR, due to decreased HDL-C, was associated with total CRAO. This observation underscores the imperative for further research into alternative blood biomarkers to facilitate the early detection of CRAO at varying levels of severity.

CRAO is widely recognised as a medical emergency that precipitates extensive acute retinal ischemia and significant visual impairment. Atherosclerosis, considered a chronic inflammatory disease, is identified as a critical risk factor for CRAO. The sclerotic retinal artery wall induces hemodynamic disturbances, destroys the vascular endothelium, and leads to thrombosis, which results in the manifestation of CRAO (Page et al., 2018).

Neutrophils, lymphocytes, and monocytes, pivotal sub-types of white blood cells, are indispensable in the body’s inflammatory response and immune defence mechanisms. The NLR and SII are recognised for their efficacy in assessing and forecasting systemic micro-inflammation, as they concurrently amalgamate two parameters (Nost et al., 2021). In this study, elevated neutrophil and monocyte counts were observed in CRAO patients compared to controls, with no significant differences noted in lymphocyte and platelet counts, resulting in higher NLR and SII values in these patients. Similar findings of significantly heightened NLR levels in CRAO patients have been documented (Guven & Kilic, 2021). CRAO is known for its association with inflammation, which exacerbates atherosclerosis by promoting endothelial thickening and luminal narrowing via inflammatory molecules and cellular pathways (Lutgens et al., 2019). Furthermore, during the progression of atherosclerosis, increased monocyte infiltration into the arterial intima augments macrophage accumulation in plaques, consequently leading to foam cell formation, characteristic of unstable plaques. Such unstable plaques, often deriving from ipsilateral carotid atherosclerosis, atrial fibrillation, or rheumatic heart valve disease, typically precipitate CRAO (Scott et al., 2020). Echoing the discoveries of Pinna et al. (2021) and Guven & Kilic (2021), our observations did not reveal a significant increase in the PLR among CRAO patients. This outcome diverges from the findings of Elbeyli et al. (2022) and Qin et al. (2022), potentially attributable to a reduction in lymphocyte counts within patients, with a definitive absence of any significant elevation in platelet numbers, suggesting that CRAO may be more intricately associated with platelet activation and volumetric increase rather than sheer quantity. Prior research posits that thrombosis is more closely correlated with platelet functionality and dimensionality than with their numerical count (Koupenova et al., 2018; Marcinkowska, Cisiecki & Rozalski, 2021).

HDL-C is renowned for its properties in reducing thrombosis. The MHR serves as a profound index that mirrors the equilibrium factors of vascular injury and protection. It is associated with various vascular conditions, such as atherosclerosis, atrial fibrillation, pulmonary embolism, acute myocardial infarction, and diabetic complications including peripheral neuropathy, nephropathy, retinopathy, and retinal branch vein occlusion (Satirtav et al., 2020; Sun et al., 2021). In the present study, an elevation in monocyte counts and a decrease in HDL-C levels were observed among patients with CRAO, resulting in an increased MHR, consistent with finding reported by Qin et al. (2022). Such an increase in MHR may suggest a predominant influence of factors related to vascular damage. This result is in stark contrast to the findings of another study (Guven & Kilic, 2021), which reported no significant changes in the MHR of CRAO patients, a variation possibly due to differences in population characteristics. To date, few studies have explored the correlation between MHR and CRAO. Given the analogous pathogenesis observed between retinal vein occlusion and CRAO, coupled with findings by Satirtav et al. (2020) that closely link MHR with branch retinal vein occlusion, the present findings lend further support the potential role of MHR in CRAO. Moreover, oxidized LDL-C, recognised as a principal risk factor for endothelial damage, exhibits substantial toxicity towards macrophages, promotes the formation of foam cells, and exacerbates atherosclerosis (Ference et al., 2017). With the progression of atherosclerosis, inflammatory cells and macrophages that engulf oxidized LDL-C are noted to release pivotal inflammatory mediators such as tumor necrosis factor-alpha, interleukin-8, and interleukin-1 (Back et al., 2019). It is notable that cholesterol emboli constitute approximately 87% of blood clots in case of CRAO, thereby underscoring CRAO as an indicator of systemic atherosclerosis (Dumitrascu, Newman & Biousse, 2020).

Different grades of CRAO are acknowledged to present specific retinal pathological changes and prognostic features (Furashova & Matthe, 2017; Matthe, Eulitz & Furashova, 2020; Schmidt, Schulte-Mönting & Schumacher, 2002). Consequently, this study was conducted to examine the variations in hematological parameters across different grades of CRAO. The findings indicated that the degree of HDL-C decline and MHR increase varied among the CRAO grades, with a notably pronounced increase in MHR observed in total CRAO. These results imply that a decrease in HDL-C is more intimately linked with the exacerbation of CRAO than variations in monocyte levels. Data recently provided by the Korean National Health Insurance Service, involving 9,316,212 participants, identified 9,878 diagnosed of CRAO over an average follow-up of 4.93 years (Hwang et al., 2022). It was observed that patients with higher HDL-C levels exhibited a reduced risk of future retinal artery occlusion, evidenced by a hazard ratio (95% CI) of 0.88 (0.83−0.95) in the fully adjusted model, thus confirming low HDL-C as an independent risk factor for retinal artery occlusion (Hwang et al., 2022). In the restricted cubic spline analysis, MHR was shown to main a statistically significant linear relationship with the occurrence and severity of CRAO, thereby emphasising its crucial regulatory role within complex networks. Additionally, the area under the curve for the combined MHR_NLR was the highest according to the receiver operating characteristic analysis, leading to the conclusion that HDL-C and MHR_NLR are critical predictors of CRAO development and valuable markers for assessing its severity.

This study has several limitations that require acknowledgement. Firstly, it is characterised by its retrospective nature and involves a small sample size, devoid of comprehensive medical information necessary for long-term follow-up. Furthermore, the assessment of additional key markers of inflammation, such as hypersensitive C-reactive protein and the albumin to C-reactive protein ratio, was not feasible. Secondly, the influence of lifestyle factors on hematological parameters, including irregular sleep patterns, work stress, alcohol consumption, and diet, could not be controlled, each of which may confound the results. Finally, the potential effects of age-related cataracts on the hematological outcomes were not considered, a factor that present considerable challengs in isolation within clinical research.

In conclusion, the study demonstrated that NLR, SII, and MHR were significantly elevated in patients with CRAO, exhibiting distinct variations in MHR levels across different CRAO grades. The highest value obtained in the receiver operating characteristic analysis for area under the curve of MHR_NLR suggests that this index may serve as an effective, economical, and convenient biomarker for assessing the severity of CRAO. While optical coherence tomography and fluorescein fundus angiography remain standard methods for diagnosing retinal artery occlusion, elevated NLR and MHR could prompt physicians and public health management centers to screen for vision loss in affected patients. Further investigation into the role of MHR_NLR in the pathogenesis of CRAO is imperative to elucidate the interplay between ocular vascular disease, immunity, and inflammation, potentially influencing the diagnosis and treatment of this condition.

Supplemental Information

Supplemental Information 1 Raw data

We acknowledge Dr. Yahan Tu for her contributions in collecting imaging data for our manuscript.

Additional Information and Declarations

Competing Interests

Author Contributions

Human Ethics

Data Availability

The authors declare there are no competing interests.

Weiwen Hu conceived and designed the experiments, performed the experiments, analyzed the data, prepared figures and/or tables, and approved the final draft.

Yikeng Huang conceived and designed the experiments, performed the experiments, analyzed the data, prepared figures and/or tables, and approved the final draft.

Qiong Zhou conceived and designed the experiments, authored or reviewed drafts of the article, fund Support, and approved the final draft.

Xionggao Huang conceived and designed the experiments, authored or reviewed drafts of the article, fund Support, and approved the final draft.

The following information was supplied relating to ethical approvals (i.e., approving body and any reference numbers):

This study was approved by the Medical Ethics Committee of the First Affiliated Hospital of Hainan Medical University on 2022087.

The following information was supplied regarding data availability:

The raw data are available in the Supplemental File.

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
