# Peer review of "Relation between peripheral blood inflammatory indices and severity of central retinal artery occlusion"

_PeerJ, doi:10.7717/peerj.18129_

## Round 0.1 · original submission · Minor Revisions

This article has now been reviewed by 2 experts, and they are recommending that some revisions are needed before it can be Accepted. Therefore, please address their comments in a revision and resubmit.

Reviewer 1 ·

Basic reporting

Q1. Introduction: The authors measured serum MHR, NLR and SII in CRAO patients to determine its correlation with the severity of CRAO. The introduction listed some related literature. However, it would benefit from a more detailed explanation of the rationale for focusing on NLR, SII, and MHR as inflammatory markers.

Experimental design

Q2. Method: Whether arteritic CRAO patients were included in the study? as the inflammatory manifestations of arteritic CRAO may affect the experimental results.
Q3. Method: The specific criteria used to classify CRAO severity into incomplete, subtotal, and total should be more clarified.
Q4. Method: It needs to be described in more detail at what time point the peripheral blood is obtained. In line 89, it is mentioned that it was obtained following a 12-hour fasting period, does this time correspond to the duration in Table 3? Or the duration in Table 3 represents the time of admission?

Validity of the findings

Q5. Results: A restricted cubic spline model could be added to further explain the relationship between CRAO and MSH and other indexes.
Q6. The small sample size included in the study may affect the persuasiveness of the article.

Additional comments

Q7. Table 2: Why are blood indexes not uniformly presented using median (IQR), and why are LDL-C and total cholesterol represented by mean values?
Q8. This article lacks clinical imaging evidence like fundus images of the CRAO group and control group (following informed consent).
Q9. minor typos: line 312: artery occlusion:: Local -> artery occlusion: Local

Reviewer 2 ·

Basic reporting

An interesting article describing relation between peripheral blood indices and CRAO. The paper is overall well-written. Please see my comments.

The beginning of the abstract should include 2-3 sentences to introduce into the topic. It would be beneficial to write more about the CRAO and its correlation with blood biomarkers, what is in this context missing.
Please write in the abstract the No. of patients in each CRAO group.
A cut-oû value of 0.124 and an area under the curve of 0.827 for MHR_NLR were found to be diagnostic tools for total CRAO - This is too specific, maybe include only in the Results section? It would be better to write exact values of NLR, SII and MHR in CRAO vs. controls.
Conclusions. Elevated levels of peripheral blood NLR, SII, and MHR are positively associated with CRAO incidence - Did you check the correlation test? If not, you cannot write like this.
Why did you assess the monocyte -to- high-density lipoprotein cholesterol ratio (MHR)? It sounds complicated.
It would be beneficial to use recent original papers to describe risk factors for CRAO in the introduction section rather than narrative reviews, especially in the field of thrombosis, thus, a following paper might be interesting to use, doi: 10.1186/s12959-023-00525-z
Why did you use the Kolmogorov-Smirnov test to assess the normality of data?
Median and Q1-Q3 ranges?
Please round p-values to the second decimal place when p>0.05, in case of p<0.05, to the third place or write p<0.001.

Experimental design

no comment

Validity of the findings

no comment

Additional comments

A pretty well discussion with in-depth analyses.

---

## Round 0.2 · accepted · Accept

Thank you for your detailed response to the two original reviewers.

Reviewer 1 ·

Basic reporting

no comment

Experimental design

no comment

Validity of the findings

no comment

Additional comments

no comment

Reviewer 2 ·

Basic reporting

Authors respond to all comments.

Experimental design

Well planned research.

Validity of the findings

Clinical useful.

Additional comments

No further issues.